# A network of epigenetic modifiers and DNA repair genes controls tissue-specific copy number alteration preference

**Dina Cramer[1,2], Luis Serrano[1,2,3]\*, Martin H Schaefer[1,2]\***

[1]EMBL/CRG Systems Biology Research Unit, Centre for Genomic Regulation, The Barcelona Institute of Science and Technology, Barcelona, Spain; [2]Universitat Pompeu Fabra, Barcelona, Spain; [3]Institució Catalana de Recerca i Estudis Avançats, Barcelona, Spain

**Abstract** Copy number alterations (CNAs) in cancer patients show a large variability in their number, length and position, but the sources of this variability are not known. CNA number and length are linked to patient survival, suggesting clinical relevance. We have identified genes that tend to be mutated in samples that have few or many CNAs, which we term CONIM genes (COpy Number Instability Modulators). CONIM proteins cluster into a densely connected subnetwork of physical interactions and many of them are epigenetic modifiers. Therefore, we investigated how the epigenome of the tissue-of-origin influences the position of CNA breakpoints and the properties of the resulting CNAs. We found that the presence of heterochromatin in the tissue-of-origin contributes to the recurrence and length of CNAs in the respective cancer type.

\*For correspondence: luis.
serrano@crg.eu (LS); martin.
schaefer@crg.eu (MHS)

**Competing interests:** The authors declare that no competing interests exist.

## Introduction

Genomic alterations in cancer show considerable heterogeneity across different tumor types and even across patients with the same type of cancer. For point mutations, we are beginning to understand the determinants of this variation: the epigenomic profile of the tissue-of-origin highly influences local mutation rates along the chromosome (*Schuster-Böckler and Lehner, 2012*; *Polak et al., 2015*; *Supek and Lehner, 2015*), different mutagens induce characteristic mutational signatures (*Alexandrov et al., 2013*), and tissue-specific exposure to environmental factors affects the selection of mutations during tumourigenesis (*Schaefer and Serrano, 2016*).

The driving forces behind copy number alterations (CNAs), that is, amplifications or deletions of genomic regions, are much less understood than the causes of point mutations. Furthermore, we do not know why some cancer types are associated with many CNAs and other types with only a few. This is partly due to the fact that CNAs tend to affect several genes at the same time [in the dataset from The Cancer Genome Atlas (TCGA; http://cancergenome.nih.gov/) used in this analysis, 59 genes on average are affected by a single CNA]. Therefore, it is often difficult to tell whether, and on which of the genes in the amplified or deleted region, selection is acting. In addition, cancer samples usually carry a much lower number of CNAs (on average 46 CNAs in the patient samples considered in this study) than single nucleotide variants (SNVs; usually 10.000s per cancer genome). The sparse number of CNAs hinders the detection of statistical associations between CNAs and genetic and epigenetic features, work that has previously been carried out for SNVs (*Schuster-Böckler and Lehner, 2012*; *Polak et al., 2015*; *Supek and Lehner, 2015*).

Like other alterations, CNAs show a large variation in position, length and number across cancer types (*Zack et al., 2013*). Authors have reported that CNA breakpoints are preferentially located in close proximity to DNA-methylation-depleted G-quadruplex sequences (*De and Michor, 2011*). This

**eLife digest** Cancer is a genetic disease that develops when a cell's DNA becomes altered. There are several different types of DNA alterations and one that is frequently seen in cancer cells is known as a "copy number alteration" (or CNA for short). These CNAs arise when breaks in the DNA are repaired incorrectly, leading to some pieces of DNA being multiplied while others are lost. Ultimately, CNAs contribute to cancer growth either by providing extra copies of genes that drive tumour development or by deleting genes that normally protect against cancer.

However, it is not known why patients with some types of cancer tend to have more CNAs than others and why some DNA regions are particularly susceptible to this type of alteration. Cramer et al. asked whether cancer patients have any other genetic mutations that might be linked with having many or few CNAs. Analysing datasets from almost 6000 patients with 20 different types of cancer showed that mutations in several genes are linked to a higher or lower number of CNAs in patients. Cramer et al. called the proteins encoded by these genes "copy number instability modulators" (or CONIMs for short).

Further investigation revealed that several of these CONIM proteins can change the way DNA is packaged inside cells. Furthermore, many of the regions of DNA that are vulnerable to CNAs in cancer cells are tightly packaged within healthy cells. These data suggest that the three-dimensional arrangement of DNA in cells influences where CNAs occur. The next step following on from this work is to find out exactly how the CONIM proteins influence the formation of CNAs.

suggests that DNA secondary structure contributes to the CNA distribution. In addition, CNAs that are close to telomeres are longer than those found in internal regions. This suggests that there are several different mechanisms of CNA generation (*Zack et al., 2013*). It has also been observed that DNA contact points in genome-wide chromosome conformation capture (HiC) proximity maps are more likely to become CNA breakpoints. Thus, the length distribution of CNAs reflects chromosomal interactions (*Fudenberg et al., 2011*). The observation that certain genes tend to be mutated in CNA-rich (TP53 and SPOP [*Ciriello et al., 2013*; *Boysen et al., 2015*]) or CNA-poor (CTCF and ARID1A [*Ciriello et al., 2013*]) cancers implies that, besides epigenetic factors, the genetic background of the cell influences CNA variation.

Here, we make use of the wealth of cancer genomics data provided by TCGA, to understand how the genetic background influences the CNA count per sample. We identify mutations in genes that are statistically linked to the number of CNAs in cancer patients. We refer to the identified gene set as CONIM genes (COpy Number Instability Modulators; *Figure 1A*). The encoded proteins form a densely interacting network of epigenetic modifiers and DNA repair genes. To test whether this network is associated with the cancer-type-specific preference for CNAs in certain regions, we investigate how the chromatin organisation in the healthy tissue-of-origin relates to the occurrence of CNAs in cancer.

## Results

### CNA number and length affect patient survival

To estimate the relevance of CNA number and length for clinical outcome, we performed Kaplan-Meier survival analyses. To this end, we grouped the patients of each cancer type into quartiles with respect to the distributions of CNA number and average length. We then compared the survival frequencies of patients in the top quartile with those of patients in the bottom quartile. It has been shown previously that cancer cells that have undergone whole genome duplications are associated with higher CNA rates (*Zack et al., 2013*) and poor prognosis (*Dewhurst et al., 2014*), thus we removed aneuploid samples. As CNA numbers have been linked to mutation rate (*Ciriello et al., 2013*), we additionally excluded highly mutated samples. We observed that for five of the 19 cancer types (brca, lgg, hnsc, paad and ucec) for which we had CNA and survival data, fewer CNAs were significantly associated with a longer survival period ($p < 0.05$; chi-square test; see *Figure 1B* as an example). In addition, in two out of the 19 cancer types (lgg and lihc), samples in the bottom quartile

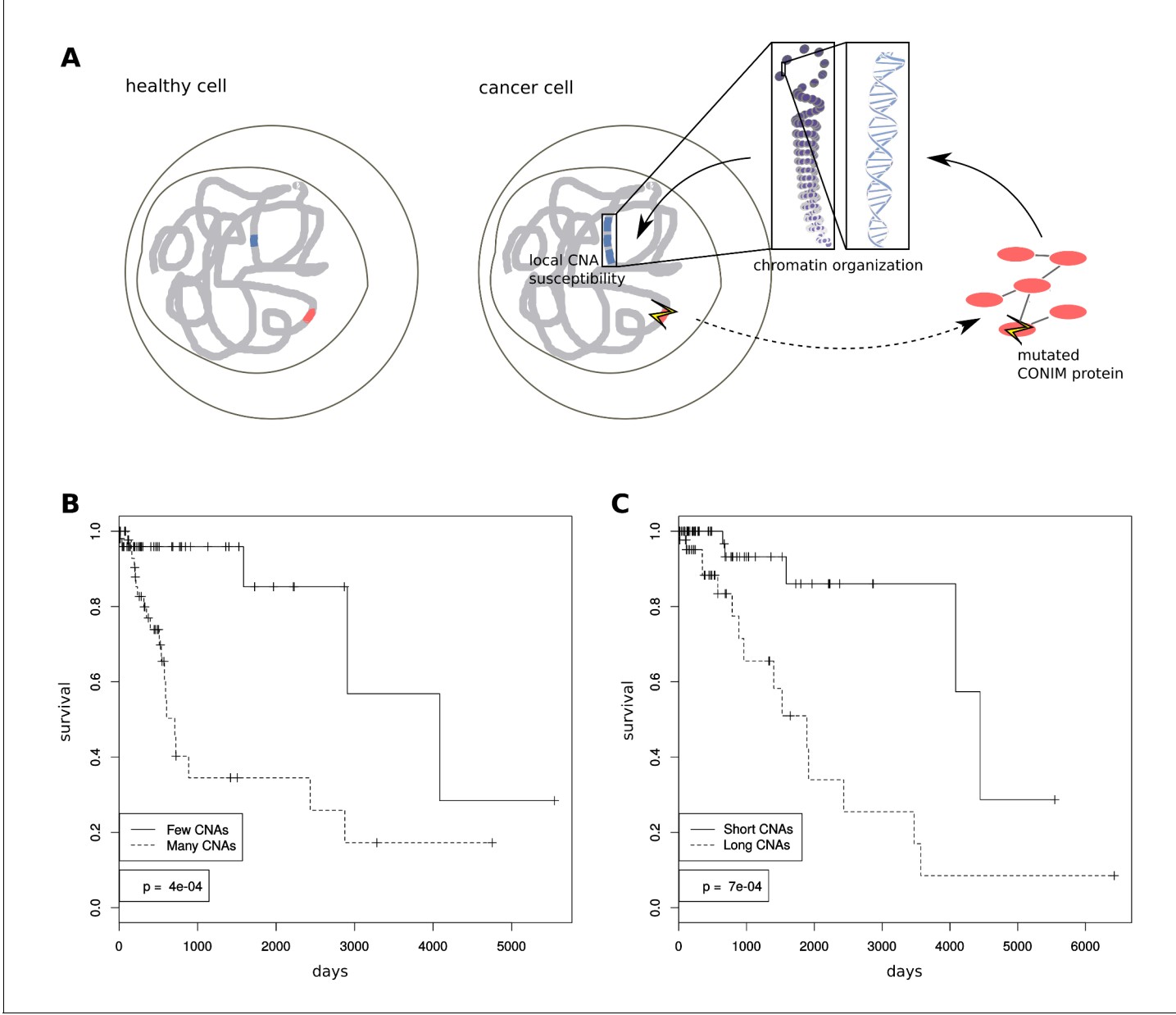

**Figure 1.** Mechanisms of CNA number modulation and clinical importance. (**A**) Schematic showing how CONIM gene mutations can result in a higher or lower CNA number. (**B**) We performed Kaplan-Meier statistics on data from lower grade glioma (LGG) patients with deviating CNA numbers and lengths. LGG patients with fewer CNAs have a significantly better survival prognosis as compared to patients with many CNAs. (**C**) LGG patients with shorter CNAs have a significantly better survival prognosis when compared to patients with longer CNAs.

of the average CNA length were associated with a longer survival compared to samples from the top quartile (p < 0.05; chi-square test; see *Figure 1C* as an example), again controlling for mutation number and ploidy. In none of the cancer types were fewer or shorter CNAs significantly associated with shorter survival.

## Gene mutations are linked to a differential CNA number

We investigated the relation between the mutational background and the CNA number of a patient. To this end, we set up a computational pipeline in order to detect genes that are associated with significantly different CNA numbers, comparing samples in which the gene was non-silently mutated

with those that were mutation-free. We corrected for potential confounding factors such as tumor-type- and gene-specific alteration rates (see Materials and methods). We applied our pipeline to a pan-cancer set consisting of 5,734 samples from 20 different cancer types (see Materials and methods). This resulted in a list of 63 genes that are associated with significantly different CNA numbers. To acknowledge the potential impact of mutations of these genes on the overall number of CNAs, we termed this gene set COpy Number Instability Modulator (CONIM) genes. Mutations in 62 of these genes are associated with significantly fewer CNAs, whereas one gene (TP53) is associated with a significantly higher number of CNAs (see *Supplementary file 1* for the full gene list and *Figure 2A* for two examples).

Of the 63 CONIM genes, 15 are known to be frequently mutated in cancer (*Lawrence et al., 2014*), and as such are likely to be drivers of malignant transformation. Their fraction among CONIM genes is higher than expected by chance (p < e-16; chi-square test). We contemplated whether mutations in the remaining 48 genes contribute to the progression of the cancer or are just a by-product of the increased mutation rates found in cancer cells. Accordingly, we used functional impact scores to estimate the pathogenicity (*Kircher et al., 2014*) of the mutations found in CONIM genes that had not been previously implicated in cancer progression. The scores were compared to those of mutations found in genes that have an equal missense mutation frequency (*Figure 2B*). We found that mutations in CONIM genes have, on average, a stronger functional impact than those in genes not associated with a change in CNA number. To estimate the temporal order of somatic events, we compared the variant allele fractions (VAFs) of mutations in non-cancer CONIM genes to the VAFs of mutations from equally often mutated genes. We found that in two out of five cancer types tested, mutations in CONIM genes were associated with a lower VAF (*Figure 2—figure supplement 1*). This suggests that mutations in CONIM genes tend to arise later in time but are more likely to be pathogenic than those in genes having similar mutation frequencies.

To investigate the potential mechanisms through which mutations in genes encoding CONIM proteins affect the amount of CNAs in a tumor, we explored the functions of the CONIM gene set. We tested for functions, pathways, and complexes enriched among CONIM genes (*Kamburov et al., 2013*). Interestingly, we found several interrelated functions to be most strongly enriched (*Figure 2C*). Among the most frequent GO terms were chromosome organisation (q < e-4; all functional enrichments FDR corrected) and chromatin modification (q < 0.001), suggesting that CONIM genes might alter CNA numbers through structural changes in the genome. More specifically, eight CONIM genes were involved in histone modification (q < 0.001). Of these, three genes were related to histone deacetylation (q < 0.01) and another three to histone methylation (q < 0.05). Together, 17 of the 63 genes had functions related to the structural organisation of the chromosomes or to epigenetic modifications (*Supplementary file 1*).

Several pathways related to DNA damage were strongly enriched [e.g., 'DNA Damage/Telomere Stress Induced Senescence' (q < 0.01) and 'DNA Damage Response (only ATM dependent)' (q < e-4)]. Notably, the ATM (Ataxia Telangiectasia Mutated) DNA damage response plays an important role in the repair of double-stranded DNA breaks from which CNAs originate. TP53 is a member of both pathways. The greater number of CNAs in TP53-mutated samples might reflect the incapability of the affected cells to repair DNA breaks or to initiate apoptosis upon damage. Another group of the most strongly enriched terms centered on complex formation [e.g. 'macromolecular complex binding' (q < e-4) and 'macromolecular complex subunit organisation' (q < 0.001)].

We tested whether we can recover the same CONIM genes when we vary the underlying CNA data or algorithmic details of the detection pipeline (see the section 'Robustness of CONIM gene discovery and properties' for details of three alternative CONIM gene detection pipelines). Even though we found that some of the CONIM genes are specifically detected by a single pipeline or in only a subset of cancer types, we found three genes that come up in all conditions and 21 genes that are recovered by at least two pipelines. Also, the enrichment of epigenetic modifiers among CONIM genes from the different pipelines is very robust.

To investigate whether CNA properties other than their number depend on the genetic background, we tested whether the average length of CNAs differs between samples with and without a mutation in each gene. We found 540 genes that are significantly associated with shorter or longer CNAs (FDR corrected q < 0.01; Mann-Whitney-Wilcoxon test). Out of these 540 genes, 122 were also associated with a significantly different CNA number (FDR corrected q < 0.01; Mann-Whitney-Wilcoxon test on pan-cancer set without applying any additional filters or corrections). The overlap

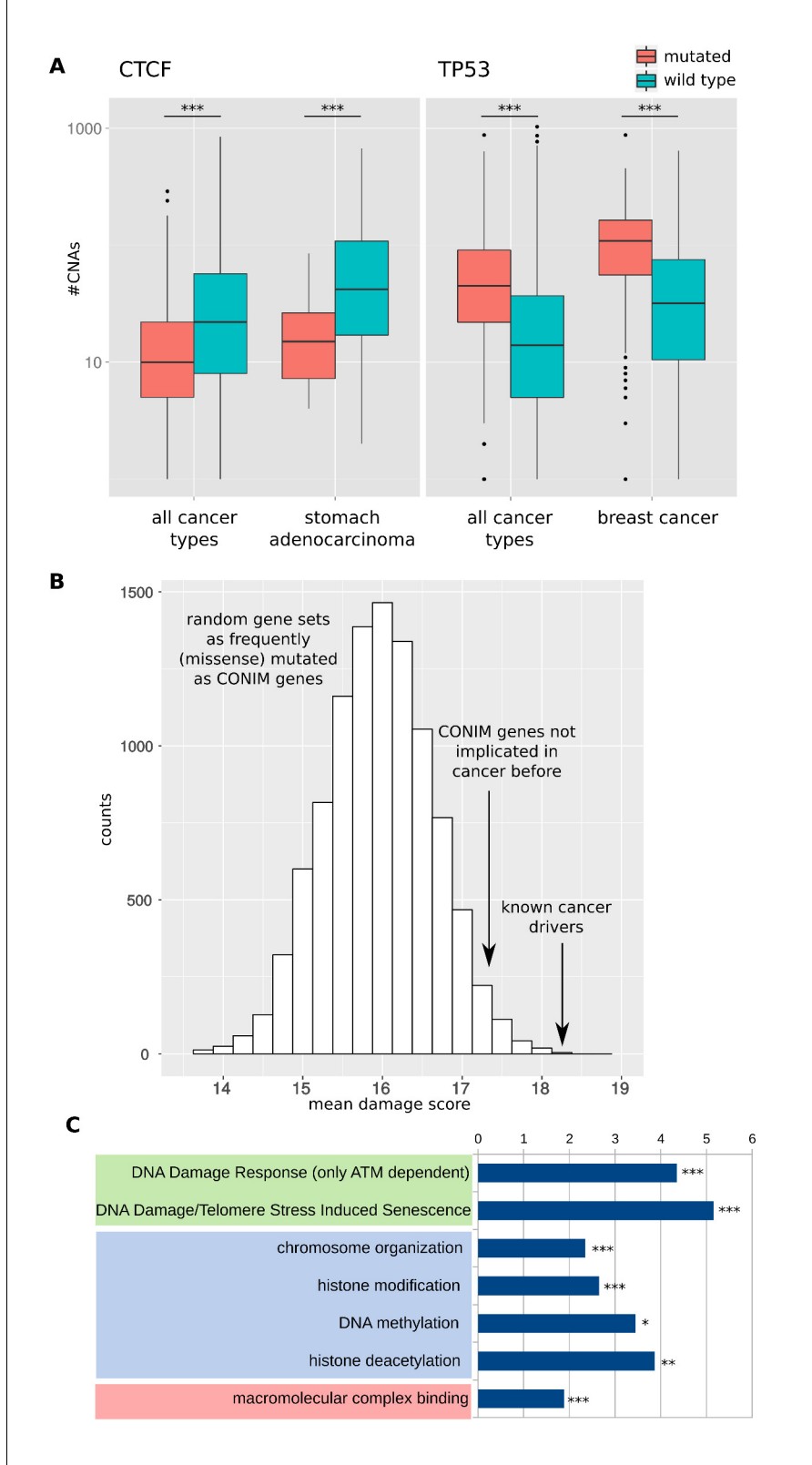

**Figure 2.** Detection and functional properties of CONIM genes. (**A**) CNA numbers in samples in which CTCF (left box) or TP53 (right box) are mutated versus samples in which the respective gene is not mutated. The CNA number distributions are shown for all cancers types (left whiskers within each box) and for a single cancer type (right whiskers within each box). (**B**) Mutations in CONIM genes tend to have a higher functional impact than mutations found in genes with an equal mutation frequency. Even CONIM genes not previously reported (*Lawrence et al., 2014*) to be frequently

*Figure 2 continued on next page*

*Figure 2 continued*

mutated in cancer tend to host mutations with a higher functional impact score (mean 17.23) as compared to random gene sets having matched mutation numbers (p = 0.029; randomisation test). For comparison, the most frequently mutated cancer-driver genes have a mean score of 18.31. (C) The functional categories most significantly overrepresented among the CONIM genes are shown. Among the most highly enriched categories are several terms related to DNA damage (green), chromatin organisation (blue) and complex formation (red). Significance levels are indicated as follows: **q < 0.01, ***q < 0.001.

The following figure supplement is available for figure 2:

**Figure supplement 1.** Variant allele fractions (VAFs) of different gene groups.

between the mutated genes found in samples with a differential CNA length and those found in samples with a differential CNA number was larger than expected by chance (p < e-16; chi-square test). The majority (98.5%) of these genes were associated with fewer and shorter CNAs, suggesting that the same cellular mechanisms might influence both CNA number and length.

## Proteins associated with a higher or lower CNA number form a dense network of interactions

As the functional enrichment analysis revealed a tendency of CONIM proteins to participate in the formation of protein complexes (*Figure 2C*), we investigated the network organisation of this protein group. When linking CONIM proteins with protein-protein interaction (PPI) information [from HIPPIE version 1.8 (*Schaefer et al., 2012*)], we observed that 32 of the CONIM proteins (50.8%) are part of a large connected network (*Figure 3A*). To test whether the degree of connectivity among CONIM proteins is greater than one would expect by chance, we performed a network randomisation test. We found that both the observed numbers of PPIs (p = 0.001; randomisation test; *Figure 3B*) and the size of the largest connected component (p = 0.003; randomisation test; *Figure 3C*) were significantly larger in the original network than in the randomised networks.

We found that CONIM proteins of the largest connected component are significantly enriched in several complexes – the four complexes with the strongest enrichment are highlighted in *Figure 3A*. In agreement with the functional enrichment, we found an enrichment of CONIM proteins in the SWI/SNF complex (EP400, ARID1A, PBRM1 and ATRX), which is involved in chromatin remodeling by restructuring nucleosomes. Mutations in components of the SWI/SNF complex have been observed in different tumor types, but their contribution to carcinogenesis is only poorly understood (*Masliah-Planchon et al., 2015*).

## Tissue-specific epigenome and chromatin organisation determine cancer-type-specific CNA breakpoint recurrence

Given previous reports on the link between chromatin structure and the genomic position of CNAs (see Introduction), we hypothesise that epigenetic modifiers are enriched among CONIM genes because they influence structural instability through chromatin modifications. In this way, CONIM genes could alter the susceptibility of chromosomal regions to DNA double-strand breaks that, when not repaired properly, would result in CNAs.

CNAs are around four orders of magnitude less abundant in patients than are SNVs. This prevented us from correlating CNA numbers from different cancer types with epigenetic marks in the respective tissue-of-origin using windows with a sufficient genomic resolution, as has been done for SNVs (*Polak et al., 2015*). Instead, we explicitly tested whether epigenetic marks around breakpoints are enriched in those tissues where the breakpoint frequently occurs during cancer development versus those tissues where the breakpoint does not occur. To this end, we assembled a list of recurrent CNAs (*Mermel et al., 2011*) that are significantly more frequent in a certain cancer type than would be expected by chance (q < 0.1; FDR corrected), resulting in 1,036 unique CNA breakpoints.

As a first analysis, we compared the frequency of 18 chromatin states (*Kundaje et al., 2015*) around the breakpoint in the tissue from which the cancer originated ('associated tissues') with the frequency in other tissues ('non-associated tissues'): *Figure 4A* shows the frequency ratios for the most abundant states. The strongest enrichment was observed for 'Heterochromatin' (p = 0.009;

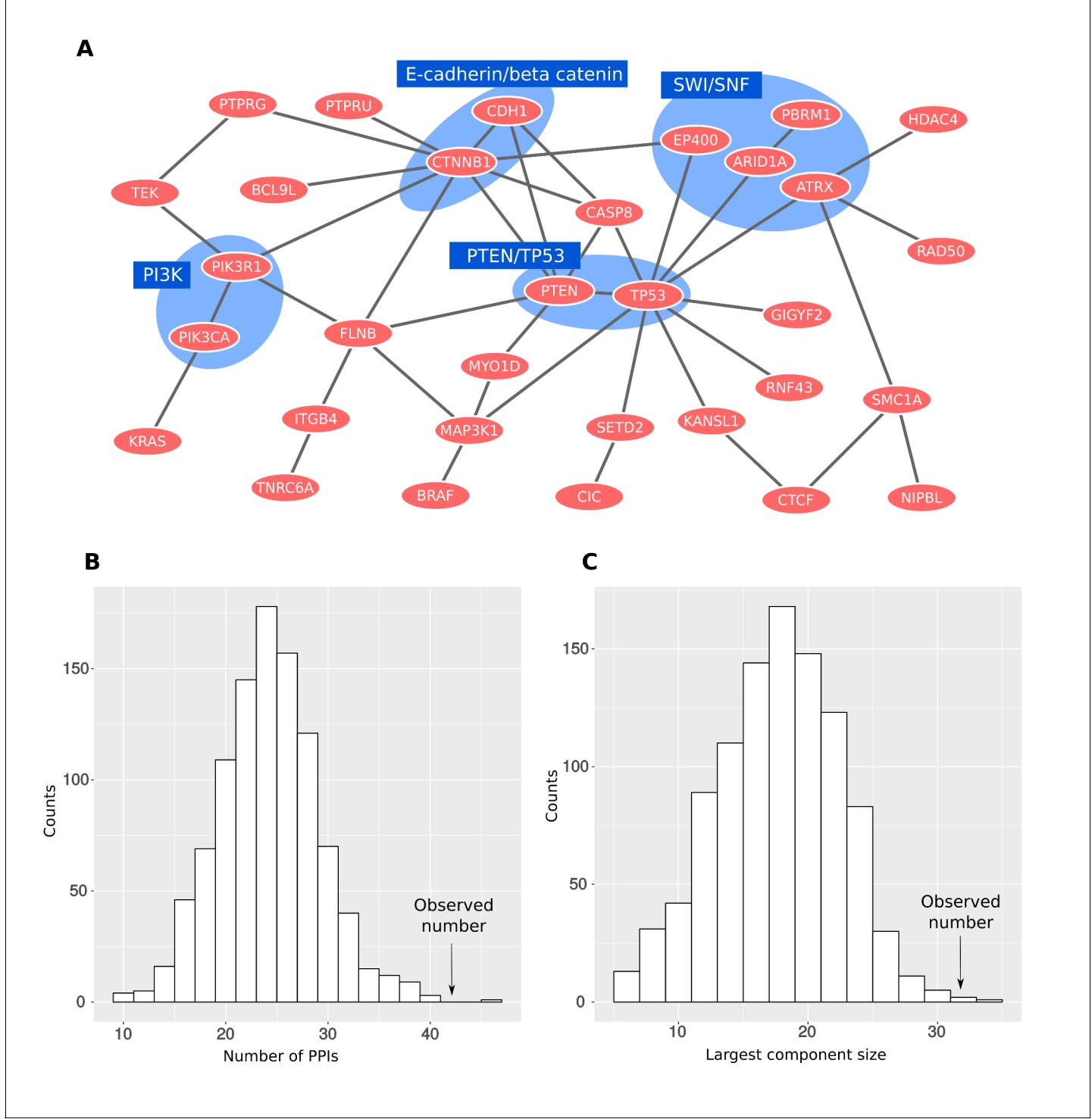

**Figure 3.** CONIM proteins form a dense network. (**A**) All interactions between CONIM proteins are shown. A total of 32 CONIM proteins are connected to each other via 42 physical interactions. Several complexes are highlighted. (**B**) The observed number of PPIs between CONIM proteins is greater than that for randomly sampled networks of proteins forming as many interactions as the CONIM proteins (p = 0.001; randomisation test). (**C**) Using the same network randomisation approach, we establish that the size of the largest connected component exceeds random expectation (p = 0.003).

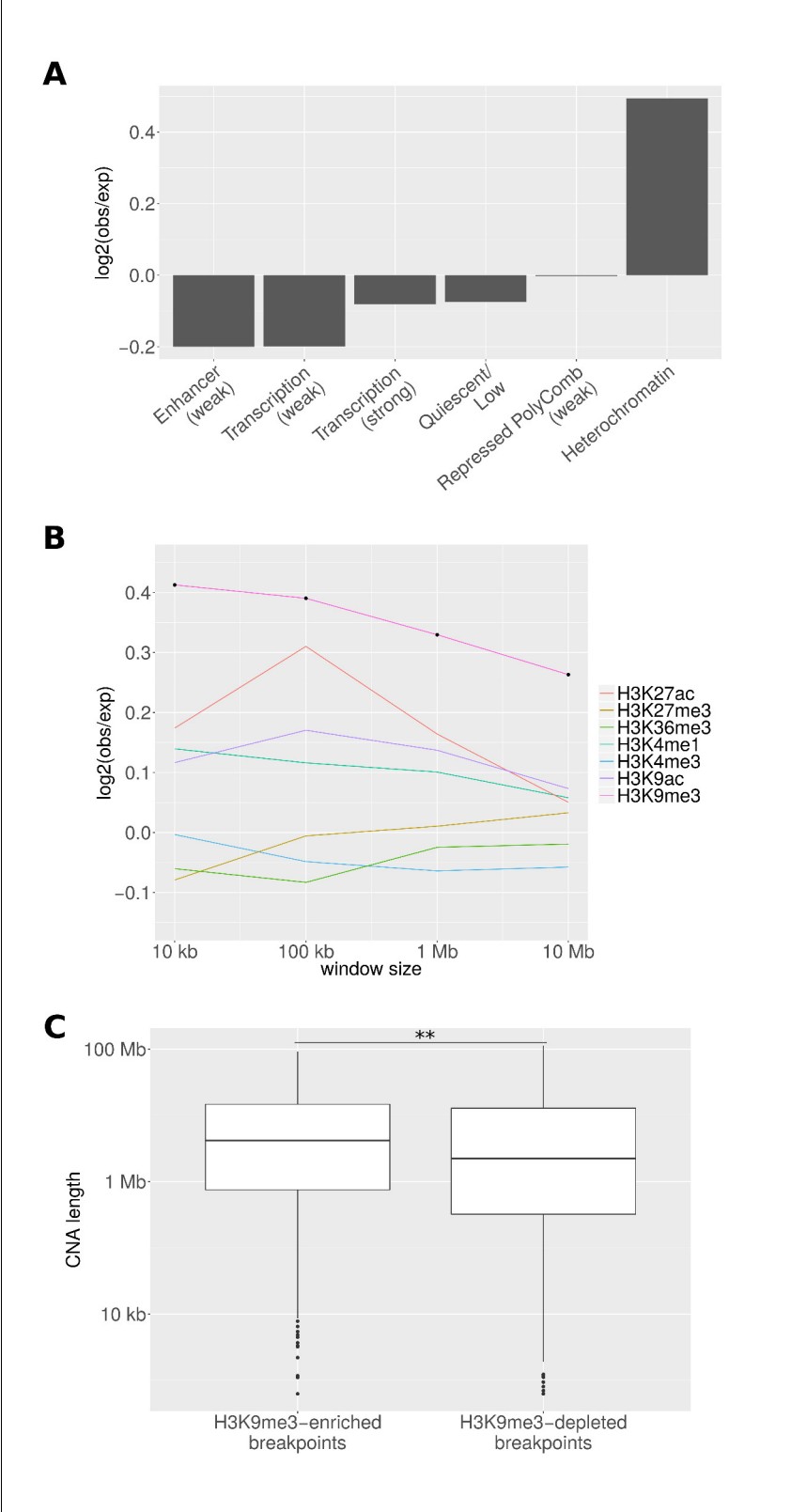

**Figure 4.** Epigenetic properties of CNA breakpoint regions. (**A**) Ratio of the number of breakpoints falling into different chromatin regions in tissues where the CNA event is significantly recurrent to the number in other tissues. States coinciding at least 100 times with breakpoints in non-associated tissues are shown. The number of CNA breakpoints in 'Heterochromatin' is significantly enriched (p = 0.009; chi-square test). (**B**) The average fraction of genomic windows centering on CNA breakpoints that is associated with different histone marks is compared between tissues where the CNA region

*Figure 4 continued on next page*

*Figure 4 continued*

drives cancer (observed) and other tissues (expected). Black dots represent bin sizes with significant enrichment (Bonferroni-corrected p < 0.05; Mann-Whitney-Wilcoxon test). (C) CNAs originating from 343 H3K9me3-enriched breakpoints are significantly longer than those originating from 738 H3K9me3-depleted breakpoints (**p < 0.01; Mann-Whitney-Wilcoxon test; 10 kb window).

The following figure supplements are available for figure 4:

**Figure supplement 1.** Enrichment of chromatin states at breakpoints for different cell-of-origin associations.

**Figure supplement 2.** Enrichment of H3K9me3 for different cell-of-origin associations.

chi-square test). The only other significantly enriched state is 'ZNF genes and repeats' (p = 0.03; chi-square test; *Figure 4—figure supplement 1*). However, the frequency of this state at CNA breakpoints is more than five times lower than that of 'Heterochromatin'.

As both of these states are characterised by the presence of H3K9me3 (*Kundaje et al., 2015*), we specifically investigated the enrichment of histone marks in the proximity of CNA breakpoints. For this purpose, we defined windows of different sizes centering on CNA breakpoints, and computed the total length of regions corresponding to a specific histone mark in tissues where the CNA region is recurrent and in other tissues. As expected, we observed the strongest enrichment for H3K9me3 (Bonferroni-corrected p < 0.001 for windows between 10 kb and 10 Mb; Mann-Whitney-Wilcoxon test; *Figure 4B*). This enrichment decreases with increasing distance from the CNA breakpoint, suggesting a colocalisation of H3K9me3 marks with recurrent breakpoints in the tissue-of-origin. For all other histone marks considered (H3K4me1, H3K4me3, H3K27me3, H3K36me3, H3K9ac, and H3K27ac), we find much weaker effects.

We next studied whether CNAs originating from H3K9me3-enriched breakpoints have any properties that distinguish them from CNAs at H3K9me3-depleted sites. As we observed the strongest H3K9me3 enrichment in 10 kb windows around CNA breakpoints (*Figure 4B*), we considered tri-methylated H3K9 in this range. We found that CNAs with a H3K9me3 enrichment in close proximity to the breakpoint were longer than CNAs originating from H3K9me3-depleted breakpoints (p = 0.001; Mann-Whitney-Wilcoxon test; *Figure 4C*). As telomere-bounded CNAs have previously been reported to be longer than others (*Zack et al., 2013*), we tested whether our observation could be an artifact of higher heterochromatin content towards the chromosome ends (even though CNAs originating from telomeres were excluded from our analyses). When comparing the positions relative to the chromosome end, we did not detect any differences between breakpoints in H3K9me3-enriched and H3K9me3-depleted genomic windows (p = 0.8; Mann-Whitney-Wilcoxon test). Additionally, we tested the effect of excluding breakpoints located within 1 Mb or 10 Mb of the chromosome ends. Both tests re-confirmed significant differences in the length distributions of the two CNA groups (p < 0.005; Mann-Whitney-Wilcoxon test). The difference in length distributions might suggest distinct mechanisms of generation that depend on the epigenetic features present at the position where the DNA breakpoint appears.

## Mechanisms of CNA generation by epigenetic dysregulation

To establish a link between tissue-specific chromatin at the CNA breakpoints and CONIM gene mutations, we sought to demonstrate that tissues with highly H3K9me3-enriched breakpoints also have more mutations in chromatin-modifying CONIM genes. None of the CONIM proteins specifically methylates H3K9, but the CONIM proteins ATRX, EP400 and NIPBL bind to H3K9me3 directly or form H3K9me3-binding complexes (*Eustermann et al., 2011*; *Lai et al., 2013*; *Oka et al., 2011*; *Vermeulen et al., 2010*; *Kunowska et al., 2015*; *Nikolov et al., 2011*). We found that non-silent mutations in these genes affect a greater proportion of samples in cancer types (luad, lusc, lihc and skcm) that show a strong H3K9me3 enrichment (> 2-fold change in 10 kb windows around breakpoints; p < 0.05; Mann-Whitney-Wilcoxon test) in their tissue-of-origin (p < e-6; chi-square test). An overrepresentation of mutated samples in these cancer types was again observed when considering each gene individually (ATRX: p = 0.02; EP400: p < 0.001; NIPBL: p = 0.09; chi-square test).

To better understand how gain- or loss-of-function mutations in CONIM genes could affect CNAs, we investigated the relationship between CONIM gene activity and heterochromatin amount

in healthy tissues. For this purpose, we compared tissue-specific RNA abundance levels (as a proxy for gene activities) with the percentage of DNA in a heterochromatic state in the same tissue. We computed the Pearson correlation between the expression of all human protein-coding genes with the percentage of heterochromatin in 48 cell lines and tissues (*Kundaje et al., 2015*). We found that the absolute correlation between total heterochromatin amount and expression of either CONIM histone modifiers or all CONIM genes is significantly larger than that of non-CONIM genes ($p < 0.05$ and $p < e-5$; Mann-Whitney-Wilcoxon test; *Figure 5A*). One possible explanation for this observation is that (under healthy conditions) CONIM genes are implicated in controlling the overall amount of heterochromatin.

We decided to focus on NIPBL, the CONIM histone modifier that showed the strongest absolute correlation ($-0.53$) between tissue-specific expression and amount of heterochromatin in the same tissue. This gene has been implicated in the developmental disorder Cornelia de Lange syndrome (CdLS) (*Krantz et al., 2004*). Mutations in NIPBL have been associated with chromatin decompaction and, indeed, mutations that are predicted to have a more severe effect on NIPBL exhibit a stronger effect on chromatin (*Nolen et al., 2013*). We therefore tested whether mutations in the HEAT domain, which is necessary to target NIPBL to sites of DNA damage (*Oka et al., 2011*), have a stronger effect on CNA number in cancers than do other missense mutations. We also checked whether cancers with truncating mutations in the N-terminus of NIPBL are associated with a significantly lower CNA number as compared to those with truncating mutations in the C-terminus (*Figure 5B*). In both cases, we observed a significant difference, with mutations that have an anticipated stronger functional or structural impact on NIPBL being associated with fewer CNAs.

These analyses suggest that the condensation state of chromatin influences the occurrence of DNA breaks. We therefore investigated whether the overall amount of heterochromatin in each tissue is linked to the amount of CNAs in the cancer type originating from the respective tissues. For most cancer types, we observed that the average number of CNAs is highly correlated to the percentage of heterochromatin in the associated tissue (*Figure 5C*). Ovarian cancer does not follow the general trend, but for other cancer types (for which we had CNA numbers and heterochromatin information available), we observed a Spearman correlation of 0.72 ($p = 0.02$). This suggests that the distribution of CNA numbers over cancer types is linked to the chromatin organisation of the tissue-of-origin.

As we found CNAs originating from breakpoints in heterochromatin to be longer, we also compared the mean length of CNAs for each cancer type with the percentage of heterochromatin in the tissue from which the cancer originated. Again, we observed a good correlation for most cancer types except for ovarian cancer (Spearman's rho = 0.85; $p = 0.002$), which decreases but remains significant when ovarian cancer is included (Spearman's rho = 0.62; $p = 0.04$; *Figure 5D*).

These observations provide a possible explanation for how mutations in CONIM genes could affect CNA numbers and lengths: the altered activity of CONIM genes affects the amount of heterochromatin, with more heterochromatin leading to more and on average longer CNAs, and with less heterochromatin having the opposite effect. The tissue-specific differences in CNA number seem to reflect the tissue-specific differences in heterochromatin.

## Discussion

Here, we describe a new class of cancer-related genes: the CONIM genes. They are characterised by being associated with the amount of chromosomal gain or loss in a cancer cell, but only about 24% of these genes have previously been associated with cancer. Our study highlights their possible role as copy number instability modulators and suggests a mechanism for how they contribute to cancer development.

Mutations in all but one of the CONIM genes are associated with a smaller number of CNAs. One explanation for this observation could be that mutations in CONIM genes tend to occur late during cancer development. This is supported by the low VAFs of CONIM genes that we observe in two cancer types. When many alterations have already been accumulated, high proliferation rates increase the risk of further damage which – at this point – would be detrimental to the cancer. The exception is TP53, which is associated with a higher number of CNAs when mutated. Inactivation of TP53 decreases sensitivity to apoptosis, and therefore more DNA damage (including CNAs) is tolerated.

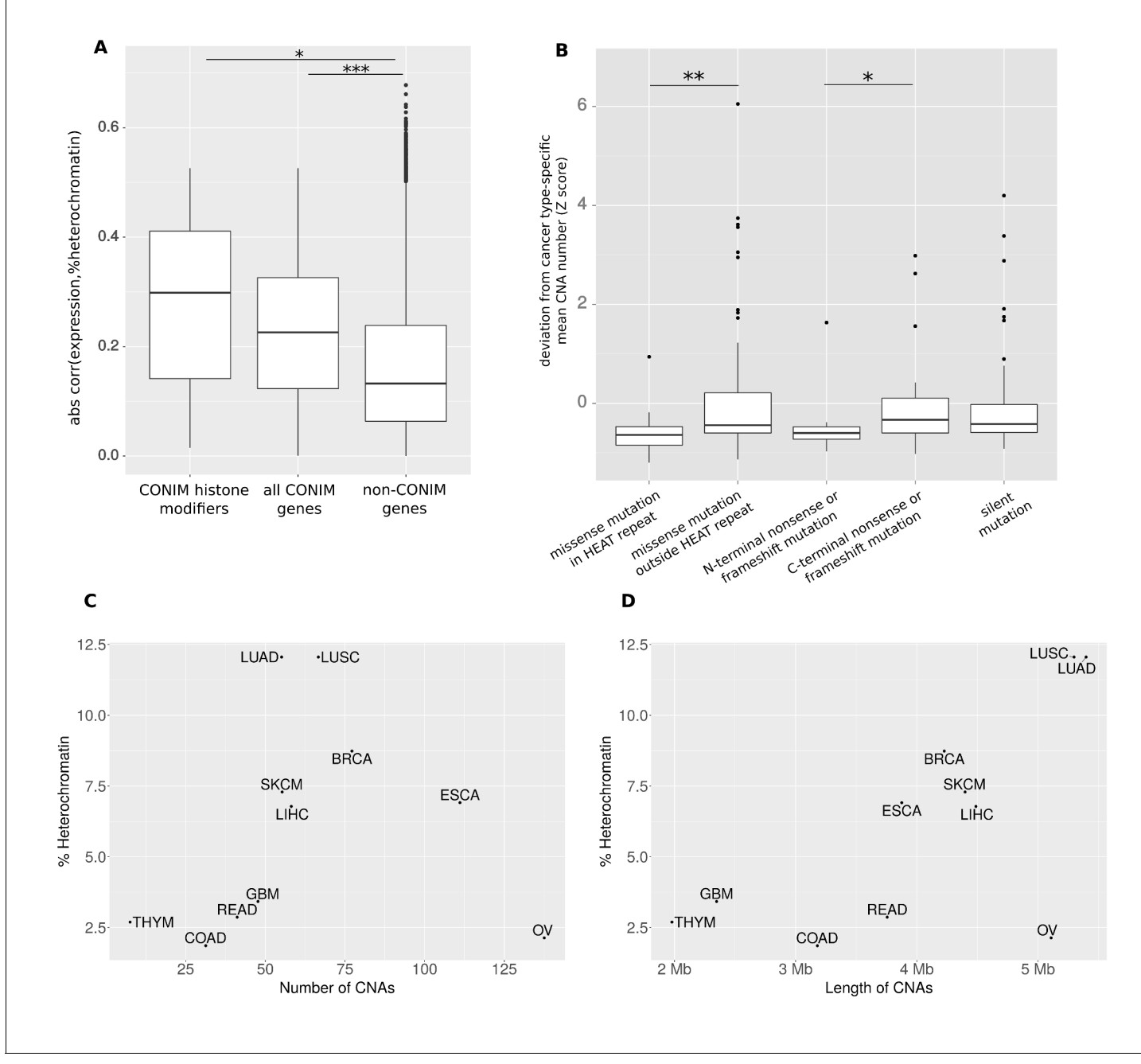

**Figure 5.** CONIM genes modify the CNA amount via the epigenome. (**A**) The absolute correlation between heterochromatin amount and expression of either CONIM histone modifiers or all CONIM genes is significantly larger than that of non-CONIM genes. (**B**) In the NIPBL gene, nonsense or frameshift mutations in the N-terminal third of the protein, and missense mutations in the HEAT repeat, have a stronger effect on the CNA number in the respective samples than do those mutations that have a smaller effect on protein structure and function. The average (**C**) CNA number and (**D**) CNA length per cancer type is correlated with the percentage of heterochromatin in the associated healthy tissue. Significance levels are indicated as follows: *: q < 0.05, **: q < 0.01, ***: q < 0.001.

The following figure supplements are available for figure 5:

**Figure supplement 1.** Average CNA number and heterochromatin percentage for alternative reference epigenomes.

**Figure supplement 2.** Average CNA length and heterochromatin percentage for alternative reference epigenomes.

Previously, an inverse relation between the number of CNAs and the number of point mutations has been described (*Ciriello et al., 2013*), subdividing tumors into two groups: one CNA-rich and one mutation-rich. The CNA-rich group has been associated with recurrent mutations in TP53 and the mutation-rich (and CNA-depleted) group with mutations in ARID1A and CTCF. These three genes are also in our CONIM gene list. Several other studies investigated relations between point mutations and CNA numbers in single cancer types: a higher number of CNAs has been reported in SPOP-mutated prostate cancer (*Boysen et al., 2015*). Lower CNA numbers have been detected in CASP8-mutated oral squamous cell carcinoma (*Pickering et al., 2013*) and in CTNNB1-mutated endometrial cancer (*Kandoth et al., 2013*). Unlike CASP8 and CTNNB1, which are part of our CONIM list, SPOP did not pass our pan-cancer CNA enrichment filter criteria because the effect of SPOP on CNAs is highly cancer-type-specific. However, SPOP was recovered by our cancer-type-specific alternative detection pipeline (see Materials and methods). Our study goes beyond these previous studies by also considering the influence on CNA occurrence of the epigenome in the tissue from which the cancer originated.

As the inverse relation between CNA and point mutations might affect the detection of CONIM genes, we apply different strategies to correct for this potential confounder (regressing out mutation rates, removing highly mutated samples and applying a mutation-number-matched permutation test). We found that the greater amount of CONIM genes associated with lower CNA number, the enrichment of epigenetic modifiers and the high connectivity can be reproduced with different CONIM gene detection pipelines. We also tested whether a gene that is associated with an elevated point mutation rate would automatically end up in our CONIM gene list due to the inverse relation between CNA and mutation counts. POLE has been described in the literature to cause a hypermutation phenotype when somatically mutated (*Roberts and Gordenin, 2014*; *Briggs and Tomlinson, 2013*). We can confirm that samples with POLE mutations have higher point mutation counts as compared to randomly selected samples (carrying mutations in genes with similar mutation frequencies as POLE). However, we do not find a reduced number of CNAs in POLE mutated samples.

The most strongly enriched pathway among CONIM genes is ATM-dependent DNA repair. ATM is required for the repair of DNA double-strand breaks in heterochromatic regions, a process which is characterised by slow repair kinetics (*Goodarzi et al., 2010*). ATM-mediated phosphorylation of KAP1 (KRAB-associated protein 1) triggers local decondensation of heterochromatin and thereby facilitates efficient repair. This suggests that it is not only the amount of cellular heterochromatin but also the cell's ability to decondense it that is important.

Other studies have begun to investigate the causes of variation in the frequency of CNAs throughout the genome by comparing distributions of CNAs to those of genomic and epigenomic features (*De and Michor, 2011*; *Fudenberg et al., 2011*; *Zack et al., 2013*). These analyses have suggested, among other features, the involvement of chromatin formation in determining the distribution of CNAs. However, none of the previous studies have systematically compared tissue-of-origin chromatin conformation to cancer-type-specific recurrence of CNAs (in a similar manner as it has been done for epigenetic marks and point mutations [*Polak et al., 2015*]). Our study complements these previous efforts by showing that not just the distribution of SNVs but also the CNA breakpoint distribution seem to be influenced by local chromatin structure.

Here, we establish a link between heterochromatin enrichment and the variation in CNA number, length, and position across cancer types. In accordance with other studies, density of chromatin (for example, differences in mechanical forces or exposure to mutagens resulting from the localisation of dense chromatin at the nuclear periphery [*Misteli, 2007*]) determines where CNAs occur or persist (the high degree of condensation might hinder the detection and repair of DNA damage [*Peterson and Côté, 2004*]). These factors are governed by the properties of the tissue-of-origin (which contribute to the variability in the number, length and distribution of CNAs over cancer types) and could be influenced by abnormal activity of epigenetic modifiers through mutation or differential expression (contributing to the variation on the patient-level). With respect to possible mechanisms of heterochromatin formation interruption, it is worth mentioning that CONIM genes encode rather more H3K9me3 'readers' than 'writers'.

Interestingly, we found that the local epigenome not only impacts where a DNA breakpoint occurs but also the length of the resulting CNAs. CNAs originating from H3K9me3-enriched regions tend to be longer than those without neighboring H3K9me3 marks. This increased average length is probably due to the fact that a greater degree of packaging of the interphase genome into

heterochromatin facilitates long-range contacts between distant parts of the DNA, which then serve as end points for CNAs. This interpretation is in agreement with the observation that the chromatin shapes the length distribution of CNAs (*Fudenberg et al., 2011*).

It has previously been reported that CNAs originating from telomeres are longer than chromosome-internal CNAs (*Zack et al., 2013*). We found that breakpoints in H3K9me3-enriched regions are associated with longer CNAs than other breakpoints, independent of their positions with respect to the chromosome ends. As telomeres in fact form heterochromatin (*Blasco, 2007*), our findings might explain the previously observed position-dependent length differences.

Regarding the link between cancer-type-specific CNA numbers or lengths and heterochromatin proportions in the corresponding tissue-of-origin, ovarian cancer may not follow the same trend as other cancer types due to very high mutation fractions in TP53 (94% [*Lawrence et al., 2014*]). In accordance with previous studies (*Ciriello et al., 2013*), we show that TP53 deficiency is strongly associated with high CNA numbers.

More research needs to be done on the mechanistic details of CNA breakpoint generation by chromatin disorganisation. To this end, our study highlights several interesting candidate genes that could be valuable drug targets as our analyses suggest that CNA number and size are clinically relevant.

In summary, our observations suggest that the epigenome impacts CNA occurrence in a tissue- and patient-specific manner. CNA breakpoints are overrepresented in heterochromatic regions, so the epigenome of the tissue from which a cancer originates has a large impact on where CNAs arise during carcinogenesis. In addition, we identified genes in which mutations are associated with differential CNA number and length. Interestingly, this gene set is enriched in chromatin-modifying genes, which could suggest that these genes influence CNA properties through chromatin modifications.

## Materials and methods

### Survival statistics

The Kaplan-Meier analysis was performed with the survival R package (https://cran.r-project.org/web/packages/survival/index.html). To prevent results from being confounded by high mutation rates, we removed samples with a mutation number of more than two standard deviations higher than the cancer-type-specific median. We also controlled for the effect of whole-genome duplications. We retrieved ploidy information for most cancer types from the COSMIC database (*Forbes et al., 2015*; RRID:SCR_002260) and the literature (*Ceccarelli et al., 2016*). For the remaining cancer types (kidney renal clear cell carcinoma and pheochromocytoma and paraganglioma), we estimated ploidy using the ABSOLUTE tool (*Carter et al., 2012*; RRID:SCR_005198). We removed samples with an estimated ploidy of more than 2.9.

### Definition of CONIM genes

We retrieved somatic mutations in coding regions for 20 cancer types [Bladder Urothelial Carcinoma (blca), Breast invasive carcinoma (brca), Cervical squamous cell carcinoma and endocervical adenocarcinoma (cesc), Colon adenocarcinoma (coad), Glioblastoma multiforme (gbm), Head and Neck squamous cell carcinoma (hnsc), Kidney renal clear cell carcinoma (kirc), Kidney renal papillary cell carcinoma (kirp), Acute Myeloid Leukemia (laml), Brain Lower Grade Glioma (gbm), Liver hepatocellular carcinoma (lihc), Lung adenocarcinoma (luad), Lung squamous cell carcinoma (lusc), Pancreatic adenocarcinoma (paad), Pheochromocytoma and Paraganglioma (pcpg), Prostate adenocarcinoma (prad), Skin Cutaneous Melanoma (skcm), Stomach adenocarcinoma (stad), Thyroid carcinoma (thac), and Uterine Corpus Endometrial Carcinoma (ucec)] from TCGA comprising a set of 5,960 samples.

CNA coordinates for each sample were retrieved from SNP6 array data through firehose (gdac.broadinstitute.org; run 2014/10/17). Only CNAs with a segment copy number larger than 0.1 or smaller than −0.1 (in units of log2(copy number) − 1) and with a minimum length of 100 bp were considered. We used 5,734 cancer samples from 20 different cancer types for which both mutation and CNA information were available, and tested whether samples with non-silent mutations in certain genes carry significantly more or less CNAs than samples without mutations in the respective

genes. We excluded genes with less than 60 non-silent mutations in the set of 5,734 samples. We only considered the cancer types that have at least 200 available samples; with at least five of the samples carrying a non-silent mutation in the respective gene. To ensure a strong effect, we only considered cases where the absolute log ratio difference was above 0.5 and applied a q-value cutoff of 0.01 (Mann-Whitney-Wilcoxon test; FDR corrected).

Among the genes associated with a higher number of CNAs, we observed a strong enrichment of those encoding large membrane-bound proteins. Among them were the largest human protein, TTN, and several olfactory receptors. As mutations in these genes are thought to be spurious passenger mutations and to confound statistics through locally elevated mutation rates (*Lawrence et al., 2013*), we additionally removed genes associated with a differential number of CNAs when carrying silent mutations. This was done by dividing samples into those carrying a silent mutation versus those not carrying a silent mutation and performing the same test as before. This implicitly corrects for local differences in mutation rates and gene length.

CNA numbers differ across cancer types and are anti-correlated with the number of mutations (*Ciriello et al., 2013*). We therefore aimed to control for these confounding factors by (a) testing in each cancer type separately for genes that when mutated are associated with higher or lower CNA number and (b) including mutation rates into a multiple regression model. For each gene$_i$ and each cancer type, we fitted a linear model with sample-specific CNA number as the predictor variable and with both mutation status of gene$_i$ and mutation number per sample as predictor variables. We then tested whether the mutation status alone significantly contributes to the CNA number (t-test). We kept only genes in the result list for which there was at least one cancer type with a FDR corrected q < 0.1.

## Properties of CONIM genes

We used the web tool ConsensusPathDB (*Kamburov et al., 2013*); RRID:SCR_002231) to assess the significance of GO term and pathway enrichment. We restricted the analysis to GO terms, pathways and complexes from the pathway databases Reactome (*Croft et al., 2014*; RRID:SCR_003485) and WikiPathways (*Kutmon et al., 2016*; RRID:SCR_002134) as well as the protein complex database CORUM (*Ruepp et al., 2008*); RRID:SCR_002254). The enrichment of all discussed functions related to epigenetic modifications and DNA repair remained significant (q < 0.05; FDR corrected) when we computed the enrichment with respect to highly mutated genes in cancer (genes with at least 100 non-silent mutations in the pooled cancer set) instead of to the entire genomic background.

We observed several genes that are involved in signaling among the CONIM set (e.g., KRAS and BRAF). However, the enrichment of signaling-related GO terms was much weaker than, for example, terms related to chromatin organisation: among the 50 most significantly enriched GO terms, none had 'signaling' but eight had 'chromosome' or 'chromatin' in the name.

The functional impact of the mutations was estimated using the Phred-transform of the CADD score (*Kircher et al., 2014*). To estimate the significance of the higher mean damage score associated with CONIM genes, a randomisation test was applied: CONIM genes not previously involved in cancer were replaced by other genes with the same number of missense mutations. We excluded genes if less than 25 other genes had exactly the same mutation count.

We computed VAFs as the read count supporting mutation divided by the total read count for each mutation in ucec, hnsc, luad, brca and skcm, as these cancer types had at least 100 mutations in non-cancer CONIM genes (considering genes with at least 15 non-silent mutations), read count information and cancer gene classification (*Lawrence et al., 2014*) available.

## Network randomisation

We retrieved PPIs from the integrated human PPI resource HIPPIE v1.8 (*Schaefer et al., 2012*; RRID: SCR_014651). To test whether the observed number of PPIs between CONIM proteins and the size of the largest connected component (the subnetwork in which every pair of proteins is connected by paths through the network) were larger than expected by chance, we performed a randomisation test. We randomly sampled 1,000 protein sets of size equal to that of the CONIM protein set by replacing each CONIM protein by a protein of the same degree (forming as many interactions as the replaced protein). This approach avoids an overestimate of connectivity statistics due to highly interacting proteins in the original protein set. In the few cases where there were less than 15 proteins

with the same interaction degree we successively increased the margin around the interaction degree of the replaced proteins until there were at least 15 proteins with the same or similar interaction degree. For each random network, we counted the size of the largest connected component and the number of PPIs formed within the random network (not considering self-interactions).

In order to exclude the possibility that the higher number of PPIs and size of the largest connected component in the original set compared to the random sets is caused by the presence of a few highly-interacting hub proteins, we removed the protein with the highest interaction degree (TP53, the only protein forming more than 500 PPIs). We then repeated the randomisation test. We again observed a larger connected component size and a higher amount of interactions than would be expected by chance (both p < 0.001; randomisation test).

## Robustness of CONIM gene discovery and properties

To estimate how robust the definition of the CONIM genes is with respect to algorithmic details and technical variation in the experimental determination of CNAs, we set up three additional pipelines. First, we designed an approach that primes cancer-type-specific CONIM genes high. To diminish the impact of variation in mutation frequency, we removed highly mutated samples (more than two standard deviations away from the median – affecting 167 samples). We then computed the significant deviation in CNA number for each cancer type separately. As before, we removed genes that show an association with CNA number when considering only silent mutations. A resulting 25 genes had a FDR corrected Mann-Whitney-Wilcoxon p-value below 0.01– and 10 of those were associated with a higher CNA number (TP53 again with the strongest effect). From the 15 genes associated with lower CNA number, 11 were found in the previous CONIM gene list. The greater proportion of CONIM genes associated with greater CNA number in this run indicates that this effect might be more cancer-type-specific.

Second, to test the degree to which the CONIM gene definition is affected by the experimental method used to detect CNAs, we retrieved another CNA set from TCGA, employing Illumina HiSeq whole-genome sequencing. This dataset is generated on a much smaller set of tumor and normal samples (856 sample pairs for which we also had mutation data), covering only 10 of the initial 20 cancer types. Applying the initial CONIM detection pipeline to this dataset revealed a smaller number of only three genes (TP53, ARID1A and PTEN) fully contained in the initial CONIM set and the cancer-type-specific CONIM gene set. However, relaxing the parameters increases the overlap with the results of the two other pipelines (for example, applying only a q-value cutoff of 0.1 on the pan-cancer CNA set results in 24 genes, nine of which are in the original CONIM set).

Third, we applied a different way of controlling for the sample-specific mutation frequencies: we permuted the observed mutations over samples and genes while keeping the number of mutations in a given gene over samples and the number of mutations in a given sample constant [following the approach described in (*Ding et al., 2008*)]. In each permutation, we computed the absolute difference in the mean CNA number between samples with and without non-silent mutations in the respective gene as a test statistic. We performed 1,000 permutations and computed an empirical p-value as the fraction of times in which the absolute CNA difference was larger than the observed difference in the original data. As before, we included only genes with at least 60 mutations in the 20 cancer types considered. For each gene, we considered only cancer types with at least five non-silent mutations in the respective gene. This resulted in a list of 48 genes that when mutated were associated with a higher or lower CNA number in the same sample (q < 0.01; permutation test). Seventeen of these genes overlapped with our initial CONIM gene definition; two of the genes were associated with higher CNA number (TP53 and OR6N1), 46 with lower CNA number.

We tested whether the genes from the alternative pipelines have the same properties as the original CONIM set: the 25 genes from the cancer-type-specific pipeline were most strongly enriched in 'DNA Damage Response (only ATM dependent)' (q < e-6). Several categories related to chromatin modification were found to be significantly enriched (q < 0.01): e.g. 'chromatin binding' and 'chromatin assembly or disassembly'. There is an enrichment of PPIs among these genes and a largest connected component exceeding random expectation (both p < 0.001; randomisation test).

Likewise, 'DNA Damage Response (only ATM dependent)' was significant (q < e-4) among the genes from the permutation-based pipeline. Also, several chromatin-modification-related categories

were enriched: e.g. 'chromatin silencing', 'chromatin modification' and 'histone methylation' (all q < 0.01). The number of PPIs formed among these genes exceeded random expectation (p < 0.05).

We did not test functional or PPI enrichment among the sequencing-based pipeline as it contains only three genes, which are fully contained in the result sets of the other three pipelines. *Supplementary file 1* gives information on the number of pipelines in which each CONIM gene can be reproduced.

## Epigenetic marks

Significantly recurrent CNAs per cancer type were retrieved from FireBrowse [firebrowse.org; SNP6 Copy number analysis (GISTIC2)] applying a q-value cutoff of 0.1. The GISTIC2 algorithm (*Mermel et al., 2011*; RRID:SCR_000151) separates arm-level and focal copy-number events, models background rates for CNA formation and defines boundaries with a predetermined confidence level.

We assigned 13 cancer types [Acute Myeloid Leukemia (laml), Breast invasive carcinoma (brca), Colon adenocarcinoma (coad), Esophageal carcinoma (esca), Glioblastoma multiforme (gbm), Liver hepatocellular carcinoma (lihc), Lung adenocarcinoma (luad), Lung squamous cell carcinoma (lusc), Ovarian serous cystadenocarcinoma (ov), Rectum adenocarcinoma (read), Skin cutaneous melanoma (skcm), Stomach adenocarcinoma (stad), Thymoma (thym)] to their tissues of origin in the Roadmap Epigenomics project (*Kundaje et al., 2015*; RRID:SCR_008924). Identifiers of selected reference epigenomes used here as well as alternative epigenomes that likewise represent potential cell types of origin are listed in *Supplementary file 2*.

We defined CNA regions as being associated with a specific healthy tissue if they were significantly recurrent in the corresponding cancer type. CNA breakpoints falling into centromere or telomere regions, as retrieved from UCSC [human genome assembly hg19 (February 2009); (*Rosenbloom et al., 2015*; RRID:SCR_005780)], and breakpoints that were associated with more than three healthy tissues were excluded from the analyses. It should be noted that the number of breakpoints for which both exclusion criteria apply is larger than expected by chance (p < e-16; Fisher's test), suggesting that most CNA breakpoints that fall into centromere or telomere regions are not tissue-specific.

For each healthy tissue, we used data from the Roadmap Epigenomics project (*Kundaje et al., 2015*) to quantify epigenetic marks for associated CNAs that are recurrent in the corresponding cancer type as compared to non-associated CNAs that promote cancer in other tissues.

We assigned a CNA breakpoint to a chromatin state if it colocalised with the genomic region corresponding to that state as defined in the 18-state model by the Roadmap Epigenomics Consortium (*Kundaje et al., 2015*). To test whether the chromatin state enrichments we observe depend on the specific reference epigenome selection, we repeated our analysis by replacing any number of reference epigenomes with equivalent cell types of origin (*Supplementary file 2*). This confirmed that the states 'ZNF genes and repeats' and 'Heterochromatin' show the most significant effects (chi-square test; *Figure 4—figure supplement 1*).

To analyze the density of histone modifications in the vicinity of CNA breakpoints, we counted the total number of base pairs that overlap with ChIP-seq peaks (ENCODE NarrowPeak format) in genomic windows centering on the breakpoint. The enrichment that we found for tri-methylated H3K9 adjacent to CNA breakpoints can be reproduced when simply counting the number of ChIP-seq peaks in a genomic window. Moreover, an enrichment of H3K9me3 can be observed for all possible cell-of-origin associations (*Supplementary file 2*; *Figure 4—figure supplement 2*; Bonferroni-corrected p < 0.005; Mann-Whitney-Wilcoxon test), suggesting that the results are independent of the reference epigenome selection.

To investigate a potential link between H3K9me3 enrichment and CNA length, we compared the length of CNAs originating from breakpoints with at least one H3K9me3 ChIP-seq peak in a 10 kb window around the breakpoint to those without neighboring H3K9me3 marks. To test whether the results of this analysis depend on the reference epigenomes that we selected, we performed this comparison for different tissue-of-origin associations (*Supplementary file 2*) and observed a significant or marginally significant difference in length distributions in all cases (p ≤ 0.05).

All results are described using GISTIC2 'region limits'. In most cases, the results hold true independent of whether 'wide peak boundaries' or 'region limits' are used to define breakpoints and independent of excluding only one or both breakpoints of CNA regions that are bounded by a genomic coordinate that falls into centromeric or telomeric regions. Exceptions are the enrichment

of the chromatin state 'ZNF genes and repeats' and the link between CNA length and H3K9me3 enrichment, where we found significant differences only when defining CNA breakpoints as GISTIC2 'region limits'.

## Heterochromatin fractions per tissues

We computed the proportion of heterochromatin from the 18-state chromatin model as defined by the Roadmap Epigenomics project (*Kundaje et al., 2015*). Likewise, we retrieved RNA expression data for protein-coding genes from Roadmap Epigenomics and we computed the Pearson correlation between the heterochromatin fraction and RNA expression for each healthy cell type for which we had RNA expression and chromatin state data.

For the analyses in which we correlated the heterochromatin fractions of tissues with CNA number and length in the corresponding cancer type, p-values testing for significance of Spearman's rho were computed with the R function cor.test, which implements the Algorithm AS 89 with Edgeworth series approximation (*Best and Roberts, 1975*). We repeated the analysis for all possible combinations of healthy tissues as well as for 1,000 random associations between heterochromatin proportions and cancer-type-specific CNA numbers and lengths. The Spearman correlation between heterochromatin percentage and CNA number (p < e-10; Mann-Whitney-Wilcoxon test; *Figure 5— figure supplement 1*) or CNA length (p < e-16; Mann-Whitney-Wilcoxon test; *Figure 5—figure supplement 2*) within the permutated reference epigenome set are significantly higher than in the random set. This holds true irrespective of whether ovarian cancer is excluded from the test statistic or not.

## Acknowledgements

We thank Stephan Ossowski and Fran Supek for helpful discussions and their comments on the manuscript, Tony Ferrar for critical manuscript revision and language editing (http://www.theeditorsite.com). The research leading to these results received funding from the German Research Foundation (SCHA 1933/1-1), the Spanish Ministry of Economy and Competitiveness, 'Centro de Excelencia Severo Ochoa 2013–2017', SEV-2012–0208, the European Union Seventh Framework Programme (FP7/ 2007–2013) under grant agreements n° HEALTH-F4-2011–278568 (PRIMES), from the Spanish Ministerio de Economía y Competitividad (Plan Nacional BIO2012-39754) and from the European Fund for Regional Development (EFRD).

## Additional information

### Funding

| Funder | Grant reference number | Author |
| --- | --- | --- |
| European Commission | HEALTH-F4-2011-278568 | Luis Serrano |
| Ministerio de Economía y Competitividad | Plan Nacional BIO2012-39754 | Luis Serrano |
| Ministerio de Economía y Competitividad | SEV-2012–0208 | Luis Serrano |
| Deutsche Forschungsgemeinschaft | SCHA 1933/1-1 | Martin H Schaefer |

The funders had no role in study design, data collection and interpretation, or the decision to submit the work for publication.

### Author contributions

DC, Analyzed the data, Wrote the manuscript; LS, Conceived the study, Wrote the manuscript; MHS, Conceived the study, Designed and coordinated the project, Analyzed the data, Wrote the manuscript

### Author ORCIDs

Martin H Schaefer, http://orcid.org/0000-0001-7503-6364

# Additional files

## Supplementary files

• Supplementary file 1. All CONIM proteins.

• Supplementary file 2. Association of cancer types to tissues of origin.

## Major datasets

The following previously published datasets were used:

| Author(s) | Year | Dataset title | Dataset URL | Database, license, and accessibility information |
|---|---|---|---|---|
| Schaefer M, Fontaine JF, Vinayagam A, Porras P, Wanker EE, Andrade-Navarro MA | 2012 | HIPPIE protein-protein interactions | http://cbdm-01.zdv.uni-mainz.de/~mschaefer/hippie/hippie_v1_8.txt | Publicly available at the Human Integrated Protein-Protein Interaction rEference website |
| Kundaje A, et al. | 2015 | Chromatin state model (18 states) | http://egg2.wustl.edu/roadmap/data/byFileType/chromhmmSegmentations/ChmmModels/core_K27ac/jointModel/final/all.mnemonics.bedFiles.tgz | Publicly available at NIH Roadmap Epigenomics Mapping Consortium |
| Kundaje A, et al. | 2015 | Histone ChIP-seq peaks | http://egg2.wustl.edu/roadmap/data/byFileType/peaks/consolidated/narrowPeak/ | Publicly available at NIH Roadmap Epigenomics Mapping Consortium |
| Kundaje A, et al. | 2015 | RNA expression (RNAseq) | http://egg2.wustl.edu/roadmap/data/byDataType/rna/expression/57epigenomes.RPKM.pc.gz | Publicly available at NIH Roadmap Epigenomics Mapping Consortium |
| Broad Institute TCGA Genome Data Analysis Center | 2014 | SNP6 focal copy number altered segments | http://gdac.broadinstitute.org/runs/analyses__2014_10_17/data/ | Publicly available at the Broad Institute website |
| Broad Institute TCGA Genome Data Analysis Center | 2015 | Illumina HiSeq copy number data | http://gdac.broadinstitute.org/runs/stddata__2015_08_21/data/ | Publicly available at the Broad Institute website |
| Broad Institute TCGA Genome Data Analysis Center | 2015 | SNP6 recurrent copy number alterations (GISTIC2) | http://gdac.broadinstitute.org/runs/analyses__2015_04_02/data/ | Publicly available at the Broad Institute website |
| Broad Institute TCGA Genome Data Analysis Center | 2014 | Mutation data | http://gdac.broadinstitute.org/runs/stddata__2014_07_15/data/ | Publicly available at the Broad Institute website |

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
