## [Decision Letter]

Thank you for submitting your article "A network of epigenetic modifiers and DNA repair genes controls tissue-specific CNA preference" for consideration by *eLife*. Your article has been reviewed by three peer reviewers, and the evaluation has been overseen by a Reviewing Editor and Aviv Regev as the Senior Editor. The following individuals involved in review of your submission have agreed to reveal their identity: Rameen Beroukhim (Reviewer #3).

The reviewers have discussed the reviews with one another and the Reviewing Editor has drafted this decision to help you prepare a revised submission.

Summary:

The focus of the paper is on identifying genes whose mutations associate with increased or decreased copy number alterations (CNAs) or changes in their lengths. The identified genes are found to be associated with epigenetic regulators, which motivated the comparison of the location of epigenetic marks with the positions of the CNA. A modestly significant enrichment for H3K9me3 and depletion of DNA methylation for the copy number alteration was then reported.

Since determinants of copy number alterations are poorly understood, the finding of this manuscript could potentially have significant impact. However the reviewers had significant concerns about the robustness of the analyses and to the extent the results were driven by confounders.

Essential revisions:

1) Of the 205 CONIM genes all but TP53 were associated with decreased mutation rates. There is concern that this is driven by confounders in particular variation in mutation rates across cancers. To quote one of the reviewer reports on this point and how this could be addressed:

"…Ciriello et al. showed that mutation and CNA rates are anticorrelated. Because high-mutation rate samples tend to have few CNAs, one would expect that most mutations would tend to be anticorrelated with CNA burden, regardless of whether there is a specific causal association. Indeed, all CONIM genes other than TP53 exhibited an association with low rates of CNAs, suggesting this confounder may be biasing the analysis. Moreover, the Methods indicate that known frequently mutated passengers (e.g. TTN) exhibited associations with CNA depletion, even when considering only silent mutations-suggesting that the results are due to confounders rather than causal relations.

To their credit, the authors recognised this as a potential issue and attempted to control for it by performing analyses within cancer types, on the supposition that CNA and mutation rates are relatively homogenous within cancer types. However, Ciriello et al. described the anticorrelation between mutation and CNA rates as occurring within cancer types as well. A better analysis would simply explicitly control for overall mutation rates when evaluating associations with CNA rates. This is easy to do and was first performed, I believe, in the Ding et al. Nature 2008 analysis of mutations in lung adenocarcinomas."

2) The authors need to better establish the results are robust to different processing pipelines they considered. Of note in the methods section an alternative pipeline with relatively small differences was applied reporting 61 genes of which only 22 were in the intersection. Of these 61, 13 were associated with higher mutations and 48 lower mutations. This raises several issues that should be addressed:

i) Since only about 10% of the 205 CONIM genes are reported by both pipelines a convincing argument needs to be made that the remaining 90% are meaningful and not driven by confounders related to differences between cancers.ii) It should be established that the results from the largest connected component in the PPI network analysis would still hold if focusing on just the set of 22 intersection genes or all 61 genes from the alternative pipeline.iii) Show the significant p-values with respect to the epigenome analysis remain significant when using genes produced from the alternative pipeline. iv) Reconcile the fraction of genes with higher mutations went from 0.5% to 21% between the two pipelines which is qualitatively different.

3) When considering the relationship between CNAs and survival time control for overall tumor ploidy. There is concern that the observed relationship could be explained by whole genome doubling, since that may allow the rate of cancer genome evolution to increase, generating a more aggressive tumor (Dewhurst & McGranahan et al., 2014).

4) When showing increased CADD scores of CONIM genes control for the step of selecting genes which were non-silently mutated. This is necessary to establish the result is not an artifact of filtering silent mutations.

5) To make the association between the enrichment of genes involved in histone methylation in the CONIM genes and the CNA breakpoints are enriched for H3K9me3 more convincing establish that those tissues with H3K9me3-enriched CNA breakpoints also have an increased number of mutations in histone methylation CONIM genes.

6) For the result that 6 of 19 cancer types fewer CNA were associated with significantly better survival and 5 of 19 with shorter CNA additional information should be reported to aid the interpretation of the results. Specifically what is the overlap between the 5 and 6 cancer types and also were there any cancer types associated with significantly worse survival?

7) Aspects of the comparison between the H3K9me3 and DNA-methylation at breakpoints should be expanded upon to enable evaluation of it. It is reported there is a significant depletion at a p-value of 0.05, but it should be stated how many actually were the intersection, which could still be substantial. When comparing H3K9me3 and DNA-methylation sites it should be clarified if only non-intersection sites were considered and stated how many sites were in each group.

8) The comparison of number of CNAs and% of heterochromatin across cancer/tissue types (Figure 6) raised questions and concerns that should be addressed. The analysis is only based on seven cancer types with one, ovarian cancer, manually excluded for being an outlier so there are concerns related to the robustness of this result. Is there are any justification for treating ovarian cancer separately? Since ovarian cancer has frequent TP53 mutations, the question was raised as to whether TP53-mutatant cancers exhibit the same relations between heterochromatin and CNA burden. It is also not clear why additional cell types with matched epigenomes listed in Table S2 and in the methods were not included in the analysis. It should be established that the relationship still holds when considering additional cell types. Also it should be clarified how in some cases the reference epigenome for the cancer type was selected when there was multiple matching ones and establish that the results are robust to that selection. For instance epigenome E061 was used for Foreskin Melanocyte Primary Cells, but there exists another reference epigenomes (E059) in the same cell type for a different individual that was not used.

9) A number of statements of causality are made in the manuscript based on correlative evidence. These statements should be adjusted to reflect their correlative nature assuming the authors do not have functional data to support them.

[Editors' note: further revisions were requested prior to acceptance, as described below.]

Thank you for resubmitting your work entitled "A network of epigenetic modifiers and DNA repair genes controls tissue-specific copy number alteration preference" for further consideration at *eLife*. Your revised article has been favorably evaluated by Aviv Regev (Senior editor), a Reviewing editor, and three reviewers.

The manuscript has been improved but there are some remaining issues that need to be addressed before acceptance, as outlined below:

The reviewers remain concerned that TP53 is the only CONIM gene not associated with decreased mutation rates and that this was not reproduced with alternative pipelines. There is concern that this result is still being driven by confounders for which the linear correction applied is inadequate. Below is a quote of one reviewer describing the issue and a better way to do the correction. It is essential that this issue be addressed.

"The manuscript is improved but the primary result-that 62 genes are associated with decreased CNA rates and only one (TP53) was associated with increased rates-raises the concern that a confounder continues to drive much of the association between mutation rates in specific genes and lack of CNAs. This may be because the authors used a linear regression model to control for varying CNA rates, whereas the relation may be non-linear and indeed appears so in Ciriello et al. The method I indicated previously should not have such an issue. This method was to control for CNA rates as per Ding et al. Nature 2008 in their analysis of lung adenocarcinomas. In this, they simply permute mutations in any gene across samples, while maintaining overall mutation rates in that sample. For instance, if one has a list of mutations as: column 1: sample id, column 2: mutated gene in that sample, one could simply permute the second column. These permutations would form a background model against which one could compare observed data to generate p-values."

A few additional points were raised that should be addressed as appropriate:

1) No CONIM genes are directly implicated in histone methylation, this should be stated in the manuscript and possible explanations discussed. In addition, the cherry-picking of a single CONIM gene for establishing a link between H3K9me3 enrichment and CONIM genes is a weakness. This portion of the manuscript could be strengthened by a similar analysis performed in a more unbiased analysis (e.g., including all genes involved in methylation "reading" instead of just EP400).

2) For the part "Out of these 540 genes, 122 were also…", these numbers were not updated to reflect the other revisions from the previous submission

3) The discussion of the VAF analysis (Discussion section, paragraph two) slightly overstates the conclusions that can be drawn from the results, since only 2 of the 5 cancer types tested had lower VAFs associated with CONIM gene mutations (subsection “Gene mutations are linked to a differential CNA number”).

4) Include some mention of the extent of agreement between alternative pipelines in results/discussion and not just methods.

---

## [Author Response]

*[…] Essential revisions:*

*1) Of the 205 CONIM genes all but TP53 were associated with decreased mutation rates. There is concern that this is driven by confounders in particular variation in mutation rates across cancers. To quote one of the reviewer reports on this point and how this could be addressed:*

"…Ciriello et al. showed that mutation and CNA rates are anticorrelated. Because high-mutation rate samples tend to have few CNAs, one would expect that most mutations would tend to be anticorrelated with CNA burden, regardless of whether there is a specific causal association. Indeed, all CONIM genes other than TP53 exhibited an association with low rates of CNAs, suggesting this confounder may be biasing the analysis. Moreover, the Methods indicate that known frequently mutated passengers (e.g. TTN) exhibited associations with CNA depletion, even when considering only silent mutations-suggesting that the results are due to confounders rather than causal relations.

*To their credit, the authors recognized this as a potential issue and attempted to control for it by performing analyses within cancer types, on the supposition that CNA and mutation rates are relatively homogenous within cancer types. However, Ciriello et al. described the anticorrelation between mutation and CNA rates as occurring within cancer types as well. A better analysis would simply explicitly control for overall mutation rates when evaluating associations with CNA rates. This is easy to do and was first performed, I believe, in the Ding et al. Nature 2008 analysis of mutations in lung adenocarcinomas."*

The reviewers and the editor are right in their assumption that the negative correlation between CNA and mutation number has an impact on the definition of CONIM genes. We are very thankful for bringing this to our awareness. In the revised version of the manuscript, we adapted our CONIM gene definition pipeline to directly control for mutation rate as a confounding factor: we modified the step where we analyze associations between mutations and CNA numbers in single cancer types by regressing mutation rates out (described in the Materials and methods section). This results in a much smaller CONIM gene list (63 genes; compared with 205 before), from which the majority (58 genes) were already part of the previous gene list. We repeated all analyses described before and found that the new CONIM gene list has similar properties as the previous list and shows an even stronger enrichment towards chromatin- and cancer-related genes:

–The fraction of known cancer genes increases from 15% to 24%.

–We find a very similar set of functional terms enriched as before (subsection “Gene mutations are linked to a differential CNA number”, Figure 2): All of the previously indicated terms related to DNA repair and chromatin modification remain significant. The fraction of genes associated with chromatin-related functions increased from 18% to 27%.

–Again, a significant fraction of proteins physically interact with each other (now 51%, before 49%; subsection “Proteins associated with a higher or lower CNA number form a dense network of interactions”, Figure 3). As only one hub gene (p53) was left in the new CONIM gene list, we repeated the analysis only removing p53 (subsection “Robustness of CONIM gene discovery and properties”). This again confirmed an enrichment of protein-protein interactions among the new CONIM set.

–The absolute correlation between expression and heterochromatin is still higher for CONIM genes (subsection “Mechanisms of CNA generation by epigenetic dysregulation”, Figure 5) than for other genes. In fact, the median absolute correlation of the CONIM genes increased by 26% (from previously 0.18 to 0.23).

–We can reproduce the results of the damage score analysis (Figure 2).

–We can reproduce the previous trend with respect to the variant allele frequency (VAF; subsection “Gene mutations are linked to a differential CNA number”, Figure 2—figure supplement 1). We adapted the analysis to the now lower CONIM gene number (e.g., going down with the minimum mutation number threshold). In the updated analysis, two out of five cancer types tested are associated with lower VAFs.

In summary, eliminating potentially false-positive CONIM genes strengthened the previously observed signals.

*2) The authors need to better establish the results are robust to different processing pipelines they considered. Of note in the methods section an alternative pipeline with relatively small differences was applied reporting 61 genes of which only 22 were in the intersection. Of these 61, 13 were associated with higher mutations and 48 lower mutations. This raises several issues that should be addressed:*

We initially did the analysis with the different pipeline to show that (a) having an association with higher CNA number is very cancer type-specific (and therefore filtering for a pan-cancer effect depletes the signal) and (b) that the functional classification is robust against the specifics of how the CONIM genes were defined. We see that in the previous version of the manuscript 1) the motivation was not very well explained, 2) presenting the two pipelines at different positions of the manuscript was confusing and 3) combining the single cancer p-values with the Fisher Method also suppresses cancer type-specific effects.

In the revised version of the manuscript, we added a new section to the Materials and methods part (“Robustness of CONIM gene discovery and properties”), in which we describe two alternative pipelines for CONIM gene discovery: one which primes cancer type-specific CONIM genes high (and does not filter for pan-cancer enrichment) and a second one to estimate the impact of technical variation (applying the same filters as for the initial CONIM gene definition to CNAs detected by whole-genome sequencing instead of SNP6 array data).

*i) Since only about 10% of the 205 CONIM genes are reported by both pipelines a convincing argument needs to be made that the remaining 90% are meaningful and not driven by confounders related to differences between cancers.*

We now recover a larger fraction of the novel primary CONIM gene list by the first alternative pipeline (19% (12 out of 63) as compared to 10% in the previous version of the manuscript). The second alternative pipeline finds only 3 of the 63 CONIM genes. However, while we observe a limited recall, the precision of the alternative pipelines is actually good: 11 of the 15 genes associated with lower CNA number in the result set of the first alternative pipeline are part of the primary CONIM gene set. Considering also genes associated with a higher CNA number according to the first alternative pipeline, the precision is 48% (12 of the 25 genes overlap with the primary CONIM set).

The second alternative pipeline reveals only three genes, which are fully contained in both the primary CONIM gene set and the gene set produced by the first alternative pipeline. The reduced amount of significant genes in the second pipeline is due to the seven-fold lower number of samples with whole-genome sequencing information, which reduces statistical power a lot.

The functional coherence among the CONIM genes and the result sets of the alternative pipelines suggests a good specificity.

*ii) It should be established that the results from the largest connected component in the PPI network analysis would still hold if focusing on just the set of 22 intersection genes or all 61 genes from the alternative pipeline.*

We repeated the analysis for the 25 genes produced by the first alternative pipeline (which fully contains the genes of the second alternative pipeline) and could reproduce the results of the network connectivity analyses.

*iii) Show the significant p-values with respect to the epigenome analysis remain significant when using genes produced from the alternative pipeline.*

The functional enrichment towards epigenetic modifiers is significant among the 25 genes of the first alternative pipeline.

*iv) Reconcile the fraction of genes with higher mutations went from 0.5% to 21% between the two pipelines which is qualitatively different.*

Our interpretation is that the association with increased CNA numbers is a more cancer type-specific effect. We now explain this idea in the Materials and methods part.

*3) When considering the relationship between CNAs and survival time control for overall tumor ploidy. There is concern that the observed relationship could be explained by whole genome doubling, since that may allow the rate of cancer genome evolution to increase, generating a more aggressive tumor (Dewhurst & McGranahan et al., 2014).*

We correct for ploidy (and mutation number – see below) in the revised version of the manuscript. Removing samples with a ploidy of 2.9 or higher has relatively little effect on our observation that higher CNA number is detrimental for survival (however, in one of the previously six cancer types, the effect falls under significance level). Removing these samples does have a stronger effect on the association between CNA length and survival: only two (of the previously five) cancer types show a significant relation.

We modified the manuscript in the Results section and in the Materials and methods part as well as Figure 1.

*4) When showing increased CADD scores of CONIM genes control for the step of selecting genes which were non-silently mutated. This is necessary to establish the result is not an artifact of filtering silent mutations.*

In the revised version of the manuscript, we replace genes from the original CONIM list with random genes carrying an equal number of missense mutations. Again, we observe a significantly higher CADD score for the CONIM gene set as compared to the random gene sets. Note that the less significant p-value is due to the higher variance in the random distribution of mean values caused by the lower number of CONIM genes while the effect size is still the same. We modified the Results and Materials and methods sections as well as Figure 2.

*5) To make the association between the enrichment of genes involved in histone methylation in the CONIM genes and the CNA breakpoints are enriched for H3K9me3 more convincing establish that those tissues with H3K9me3-enriched CNA breakpoints also have an increased number of mutations in histone methylation CONIM genes.*

The CONIM genes involved in histone methylation are not directly implicated in H3K9 trimethylation. Instead, we included the suggested analysis taking the example of EP400, which is part of the H3K9me3-binding Tip60 histone acetyltransferase complex. We added a paragraph to the Results section.

*6) For the result that 6 of 19 cancer types fewer CNA were associated with significantly better survival and 5 of 19 with shorter CNA additional information should be reported to aid the interpretation of the results. Specifically what is the overlap between the 5 and 6 cancer types and also were there any cancer types associated with significantly worse survival?*

We now indicate the respective cancer types in the manuscript, and that there was not a single cancer type with a significant association to beneficial outcome for higher number or length of CNAs (subsection “CNA number and length affect patient survival”).

*7) Aspects of the comparison between the H3K9me3 and DNA-methylation at breakpoints should be expanded upon to enable evaluation of it. It is reported there is a significant depletion at a p-value of 0.05, but it should be stated how many actually were the intersection, which could still be substantial. When comparing H3K9me3 and DNA-methylation sites it should be clarified if only non-intersection sites were considered and stated how many sites were in each group.*

The information about the number of H3K9me3-enriched and DNA methylation-depleted breakpoints can be found in the Venn diagram in Figure 5 of the previous version of the manuscript. For the comparison of CNA lengths, we considered non-intersection sites only. However, we replaced this analysis in the revised version of the manuscript by a test that is based on H3K9me3 presence only. We now indicate more clearly the number of sites involved in the test (Figure legend 4). We visualize the test results in Figure 4.

*8) The comparison of number of CNAs and% of heterochromatin across cancer/tissue types (Figure 6) raised questions and concerns that should be addressed. The analysis is only based on seven cancer types with one, ovarian cancer, manually excluded for being an outlier so there are concerns related to the robustness of this result. Is there are any justification for treating ovarian cancer separately? Since ovarian cancer has frequent TP53 mutations, the question was raised as to whether TP53-mutatant cancers exhibit the same relations between heterochromatin and CNA burden. It is also not clear why additional cell types with matched epigenomes listed in Table S2 and in the methods were not included in the analysis. It should be established that the relationship still holds when considering additional cell types. Also it should be clarified how in some cases the reference epigenome for the cancer type was selected when there was multiple matching ones and establish that the results are robust to that selection. For instance epigenome E061 was used for Foreskin Melanocyte Primary Cells, but there exists another reference epigenomes (E059) in the same cell type for a different individual that was not used.*

When analyzing the relation between heterochromatin proportions and CNA numbers or lengths, we now include cancer types with less than 200 samples (esca, read, thym) and luad as a second matched cancer type for lung tissue.

We agree with the reviewers that ovarian cancer may not show the same behavior like other cancer types due to frequent TP53 mutations and we include a corresponding paragraph in the Discussion section.

In the revised version of the manuscript, we report that CNA length is significantly correlated with heterochromatin percentage independent of including or excluding ovarian cancer.

Our reference epigenome selection is based on better quality control statistics (E061 instead of E059) or lower age of donors (E101 instead of E102) if data corresponding to the same cell type is available for more than one individual (Kundaje et al. Nature. 2015). In the revised version of the manuscript, we include alternative matched epigenomes in [Supplementary-material SD1-data]. We repeat all analyses in which Roadmap Epigenomics data is used with different reference epigenome selections (Materials and methods, Figure 4—figure supplement 1 and Figure 4—figure supplement 2, Figure 5—figure supplement 1 and Figure 5—figure supplement 2) and thereby corroborate our results.

We thank the reviewers and editor for the suggestion to use alternative associations between cancer patients and healthy epigenomes to test the robustness of our analyses.

*9) A number of statements of causality are made in the manuscript based on correlative evidence. These statements should be adjusted to reflect their correlative nature assuming the authors do not have functional data to support them.*

Throughout the manuscript, we modified or removed formulations, which implied directionality or causality based on correlative evidence. This affects the relation between CNA properties and survival, between mutations and CNA number and between heterochromatin and gene expression. Where we left them, we stated more clearly that the assumption of causality is a hypothesis.

[Editors' note: further revisions were requested prior to acceptance, as described below.]

*[…] The reviewers remain concerned that TP53 is the only CONIM gene not associated with decreased mutation rates and that this was not reproduced with alternative pipelines. There is concern that this result is still being driven by confounders for which the linear correction applied is inadequate. Below is a quote of one reviewer describing the issue and a better way to do the correction. It is essential that this issue be addressed.*

*"The manuscript is improved but the primary result-that 62 genes are associated with decreased CNA rates and only one (TP53) was associated with increased rates-raises the concern that a confounder continues to drive much of the association between mutation rates in specific genes and lack of CNAs. This may be because the authors used a linear regression model to control for varying CNA rates, whereas the relation may be non-linear and indeed appears so in Ciriello et al. The method I indicated previously should not have such an issue. This method was to control for CNA rates as per Ding et al. Nature 2008 in their analysis of lung adenocarcinomas. In this, they simply permute mutations in any gene across samples, while maintaining overall mutation rates in that sample. For instance, if one has a list of mutations as: column 1: sample id, column 2: mutated gene in that sample, one could simply permute the second column. These permutations would form a background model against which one could compare observed data to generate p-values."*

We agree with the reviewers and the editor that taking the variation of mutation rates over samples into account is important as it might impact our results. We performed the analysis suggested by the reviewer and a few other tests to demonstrate that our observations hold.

We applied the permutation test suggested by the reviewer (as described by Ding et al. 2008). We observed that even with this stringent correction for sample-specific mutation rates the ratio between genes associated with high CNA number versus genes associated with low CNA numbers is similar as in the previous CONIM gene list (2 out of 48 are associated with higher CNA number – including, again, TP53). 17 of the these 48 genes have been part of the original CONIM list. As in the initial list, we observe a strong enrichment towards DNA repair genes and epigenetic modifiers. Even among the non-overlapping genes (not in the previous CONIM set), we find more histone modifiers than expected by chance. This suggests that even though we have a variation in dependence of the specific algorithmic details or the underlying data, we observe a robust functional enrichment towards epigenetic modifiers and a higher number of genes associated with lower CNA number (with exception of the cancer type-specific pipeline where the second effect is less pronounced).

In the revised version of the manuscript, we added a description of the permutation test to the Materials and methods part (in the “Robustness of CONIM gene discovery and properties” section). We also simplified the same section by removing the discussion of the functional properties of the overlap between the cancer type-specific pipeline with the CONIM genes. Instead we show the results of the functional and PPI enrichment analysis for all result sets from the different pipelines separately. To better illustrate the overlap between the different CONIM gene detection pipelines and to make the highly reproducible core CONIM set more easily accessible, we now indicate in [Supplementary-material SD1-data] in how many of the alternative pipelines a CONIM gene was identified. We also extended the discussion adding a new section.

Author response image 1.**DOI:**
http://dx.doi.org/10.7554/eLife.16519.015

Above we illustrate the agreement of the alternative pipelines with the original CONIM set (pipeline 2 = cancer type-specific pipeline; pipeline 3 = illumina whole genome sequencing data; pipeline 4 = permutation-based approach as suggested by the reviewer). We decided to stay with the previous pipeline given the overlap between the previous pipeline and the one suggested by the reviewer. Also, they share similar properties with respect to functional and interaction enrichment as well as the ratio between genes associated with higher and those associated with lower CNA counts. Instead, we discuss the different pipelines and sources of the variability between them in the Discussion section. However, if the reviewer insists we would be willing to change the manuscript and present ours and the permutation-based approach as the main pipelines.

*A few additional points were raised that should be addressed as appropriate:*

*1) No CONIM genes are directly implicated in histone methylation, this should be stated in the manuscript and possible explanations discussed. In addition, the cherry-picking of a single CONIM gene for establishing a link between H3K9me3 enrichment and CONIM genes is a weakness. This portion of the manuscript could be strengthened by a similar analysis performed in a more unbiased analysis (e.g., including all genes involved in methylation "reading" instead of just EP400).*

In the revised version of the manuscript, we state that no CONIM protein is directly involved in H3K9 methylation but rather in reading H3K9me3 in the Results and Discussion sections. As suggested by the reviewer, we systematically searched for H3K9me3 “readers” in the literature and analyzed the genes ATRX and NIPBL in addition to EP400. We again observed elevated mutation frequencies in cancer types that correspond to H3K9me3-enriched tissues. We moved the paragraph describing this analysis to the Results section “Mechanisms of CNA generation by epigenetic dysregulation”.

*2) For the part "Out of these 540 genes, 122 were also…", these numbers were not updated to reflect the other revisions from the previous submission*

In fact, the numbers still hold as in this comparison we do not correct for mutation frequencies. Also, we do not filter genes that do not give a signal in single cancer types or those that are also associated with silent mutations. We just want to illustrate with this comparison that CNA number and length are not independent. However, we agree that it is confusing that the overlap is higher than the previously stated total number of CONIM genes. We therefore clarify that we did not filter at the respective position in the revised manuscript.

*3) The discussion of the VAF analysis (Discussion section, paragraph two) slightly overstates the conclusions that can be drawn from the results, since only 2 of the 5 cancer types tested had lower VAFs associated with CONIM gene mutations (subsection “Gene mutations are linked to a differential CNA number”).*

Yes, we agree. We reformulated the respective sentences in the Results and Discussion sections to mark this conclusion more clearly as a hypothesis. We also updated the corresponding Figure 2—figure supplement 1 where one of the gene boxes was erroneously plotted twice.

4) Include some mention of the extent of agreement between alternative pipelines in results/discussion and not just methods.

We added paragraphs to the Result and Discussion sections, in which we mention and discuss the overlap between the different pipelines. We also restructured the Discussion section to be easier to follow.